# Learning Fast-Inference Bayesian Networks

**Vaidyanathan Peruvemba Ramaswamy**
Algorithms and Complexity Group
TU Wien, Vienna, Austria
vaidyanathan@ac.tuwien.ac.at

**Stefan Szeider**
Algorithms and Complexity Group
TU Wien, Vienna, Austria
sz@ac.tuwien.ac.at

## Abstract

We propose new methods for learning Bayesian networks (BNs) that reliably support fast inference. We utilize *maximum state space size* as a more fine-grained measure for the BN's reasoning complexity than the standard treewidth measure, thereby accommodating the possibility that variables range over domains of different sizes. Our methods combine heuristic BN structure learning algorithms with the recently introduced MaxSAT-powered local improvement method (Peruvemba Ramaswamy and Szeider, AAAI'21). Our experiments show that our new learning methods produce BNs that support significantly faster exact probabilistic inference than BNs learned with treewidth bounds.

## 1 Introduction

The time complexity of probabilistic reasoning on a Bayesian Network (BN) is dominated by the *maximum state space size* of clusters (i.e., bags of the BN's tree decomposition [Lauritzen and Spiegelhalter, 1988, Dechter, 1999, Kask et al., 2011]; a bag's state space size is the product of the domain sizes of all the variables it contains). We propose algorithms for learning BNs from data, keeping the state space size within a user-specified bound. This results in *fast-inference* BNs, i.e., BNs reliably admitting fast probabilistic reasoning. We compare our algorithms to the baseline of state-of-the-art bounded treewidth BN learning algorithms, on real-world benchmark data sets, with up to over a thousand variables. The results show a clear advantage for new bounded state space (*bss*) algorithms.

It is common to encounter non-binary variables in real-world data. Moreover, during our preliminary analysis, we noticed that even variables with domain sizes as small as 4 were sufficient to impact the reasoning times significantly. This is in agreement with the fact that the reasoning time has an exponential dependence on the domain sizes. For instance, consider some of the networks learned for `alarm` and `hepar2` having 37 and 70 variables, respectively. Both these datasets were learned with small values of treewidth, and the maximum domain size of the variables is 4. Despite this, they exhibited reasoning times in the order of magnitude of 2.5 seconds.

For our bss BN structure learning algorithms, we build upon recent work on bounded treewidth BN structure learning, particularly on the heuristic algorithms k-greedy and k-MAX by Scanagatta et al. [2016, 2018] as well BN-SLIM by Peruvemba Ramaswamy and Szeider [2021a]. The latter is a post-processing algorithm that uses MaxSAT to improve BNs generated by the heuristics. All these algorithms assume a user-specified upper bound $k$ for the treewidth of the learned BN and optimize the BN's score under the given treewidth bound. The learning algorithms are highly optimized for dealing with large instances, and so the generalization of treewidth bounds to state space bounds isn't straightforward. The main challenge for extending BN-SLIM to bss learning is to replace BN-SLIM's simple cardinality constraints with a MaxSAT encoding that bounds the state-space of a bag, i.e., a product of integers. We achieve this by switching to logarithms and bounding the sum of real numbers, utilizing a MaxSAT encoding based on *binary decision diagrams (BDDs)*.

35th Conference on Neural Information Processing Systems (NeurIPS 2021).

We consider several variants of bss learning algorithms, tested them on 16 real-world benchmark data sets with up to 1041 variables, and compare them with bounded treewidth BN learning algorithms. For the comparison, we put pairs of scatter plots side by side, which show the tradeoff between reasoning speed and data-fitting (score), one for the baseline methods and one for the bounded state space methods. The bounded state space methods show better performance throughout, with significantly higher reliability (small variance).

## 1.1 Related Work

We discuss related work in terms of Figure 1. In approaches (a) and (b), the BN has already been

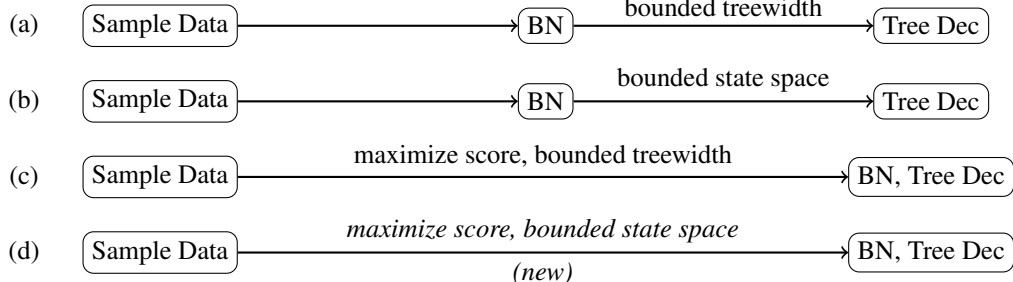

Figure 1: Various approaches to BN structure learning

learned by some other method, and one tries to find a tree decomposition that minimizes either the treewidth or the maximum state space size, respectively. For approach (a), general-purpose tree decomposition algorithms can be applied, such as the one by Gogate and Dechter [2004]. As the significance of the state space of BNs was recognized [Lauritzen and Spiegelhalter, 1988], the research focused on approach (b) [Kask et al., 2011, Kærulff, 1992, Meila and Jordan, 1996, Otten and Dechter, 2008a,b]. However, once the BN has been fixed, the impact of the decomposition method is limited. Therefore, the research in the last decade focused on approach (c), where a treewidth bound is already considered during the BN learning; a suitable decomposition is produced simultaneously. On the one hand, exact learning algorithms have been proposed that scale only to small instances but find score-optimal BNs [Korhonen and Parviainen, 2013, Parviainen et al., 2014, Berg et al., 2014]. On the other hand, heuristic algorithms have been proposed that scale to large instances but do not guarantee score optimality [Nie et al., 2015, 2016, Scanagatta et al., 2016, 2018, Benjumeda et al., 2019]. Recently, Peruvemba Ramaswamy and Szeider [2021a] proposed the hybrid approach BN-SLIM, which improves the score of a heuristically computed BN by multiple applications of a MaxSAT-based exact method.

In this paper, we follow for the first time approach (d) and implement it via various scalable algorithms. Our experiments compare approaches (c) and (d) in terms of the achieved score and inference speed.

## 2 Preliminaries

### 2.1 Graphs and Digraphs

We consider both undirected and directed graphs $G$, where $V(G)$ denotes the set of vertices and $E(G)$ the set of edges of $G$, respectively. If $G$ is directed, we refer to its edges as arcs. A directed graph is a DAG if it doesn't contain any directed cycles.

### 2.2 Structure Learning

Assume we are given a data set of samples $D_1, \ldots, D_N$ over $n$ categorical random variables, each variable $v$ ranging over a domain of $\mathrm{ds}(v)$ many discrete values. The goal of the BN structure learning problem is to find a DAG $D$, whose vertices are the random variables, that fits the data best. The fitting is modeled employing a real-valued *score function*. The BN is formed by the DAG $D$ and local parameters [Koller and Friedman, 2009]. More precisely, the score function $f$ assigns each node $v \in V(D)$ and each subset $P \subseteq V \setminus \{v\}$ the *score* $f_P(v)$ of $P$ for $v$. Let

$P_D(v) = \{\, u \in V : (u, v) \in E \,\}$ denote the parent set of $v$ in $D$. The score $f(D)$ of $D$ is the sum of $f(v, P_D(v))$ over all $v \in V(D)$. Score functions defined in such a way (decomposable score functions) accommodate popular scores like AIC, BDeu, and BIC [Akaike, 1974, Heckerman et al., 1995, Schwarz, 1978]. A *score function cache* is obtained from a score function by disregarding any potential parent set $P'$ of a node $v$ if $f(v, P') \leq f(v, P)$ for another potential parent sets of $v$ with $P \subsetneq P'$. Consequently, a score function cache does not consider a nonempty potential parent set if its score doesn't exceed the score of the empty parent set.

### 2.3 Tree Decomposition

Consider an undirected graph $G$. A *tree decomposition* $\mathcal{T}$ of $G$ is a pair $(T, \chi)$, consisting of a tree $T$ and a mapping $\chi : V(T) \to 2^{V(G)}$, where the set $\chi(t)$ is called a *bag*, with the following properties.

1. For every edge $e \in E(G)$ there is some $t \in V(T)$ such that $e \subseteq \chi(t)$.
2. For every vertex $v \in V(G)$, the set $\{\, t \in V(T) : v \in \chi(t) \,\}$ induces a non-empty subtree of $T$.

The *moralized graph* $M(D)$ of a DAG $D$ is defined by $V(M(D)) = V(D)$ and $E(M(D)) = \{\, \{u, v\} : (u, v) \in E(D) \,\} \cup \{\, \{u, v\} : (u, w), (v, w) \in E(D), u \neq v \,\}$. A tree decomposition of a DAG is a tree decomposition of its moralized graph. A tree decomposition of a BN is a tree decomposition of its underlying DAG.

## 3 Treewidth and Maximum State Space Size

In this section, we discuss the different metrics that can be used to estimate the inference speed of a BN along with some empirical findings.

Consider a tree decomposition $\mathcal{T} = (T, \chi)$ of a DAG $D$, where $V(D)$ consists of random variables, each $v \in V(D)$ ranging over a set of $\mathrm{ds}(v)$ many discrete values.

The *width* of $\mathcal{T}$ is

$$\max_{t \in V(T)} |\chi(t)| - 1,$$

i.e., is the size of a largest bag minus 1. The *treewidth* $\mathrm{tw}(D)$ of $D$ is the minimum width over all tree decompositions of $M(D)$.

The *maximum state space size* of $\mathcal{T}$ is

$$\max_{t \in V(T)} \prod_{v \in \chi(t)} \mathrm{ds}(v),$$

i.e., largest state space size of all bags, where the state space of a bag is the product of the domain sizes of the variables it contains. The *maximum state space size* $\mathrm{msss}(D)$ of a $D$ is the minimum msss over all tree decompositions of $M(D)$.

The maximum state space of a *binary* BN of treewidth $t$ is $2^{t+1}$. The *bounded treewidth (state space, respectively) BN structure learning problem* takes as input a set $V$ of nodes (i.e., random variables), a decomposable score function $f$ on $V$, and an integer $k$, and asks to compute a DAG $D$ with $V(D) = V$ of treewidth (maximum state space size, respectively) at most $k$, with a maximal score $f(D)$.

### 3.1 Empirical Influence on Inference Speed

Complexity results for probabilistic reasoning (inference) suggest that for BNs containing non-binary variables, maximum state space size provides a more accurate prediction for inference speed than treewidth [Lauritzen and Spiegelhalter, 1988, Dechter, 1999, Kask et al., 2011]. Our initial experiments aimed to verify this theoretical assumption empirically. For this purpose, we generated several BNs with varying treewidth and maximum state space size and analyzed the impact of these two measures on the BN's inference speed. We define a BN's reasoning time as the time required for computing the probability of evidence of 5 random variables set to random states, averaged over 100 runs (same as [Scanagatta et al., 2016]). For more details on the experimental setup, we refer to Section 5.

Figure 2 depicts the distribution of the observed reasoning times for different treewidth and maximum state space size ranges utilizing boxplots, with dots signifying outliers. We observe that the correlation

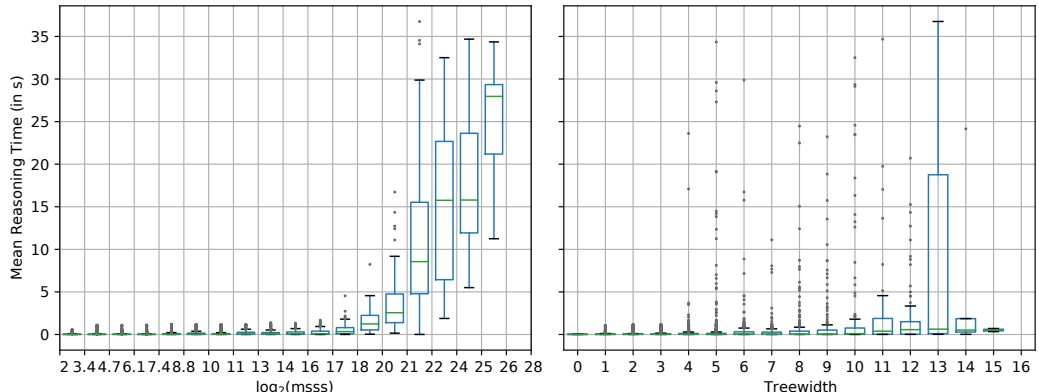

Figure 2: Comparison of correlation of treewidth and maximum state space size with reasoning time

between reasoning time and maximum state space size is much stronger than the correlation between reasoning time and treewidth. The number of outliers is much higher in the treewidth plot as compared to the maximum state space size plot. Furthermore, for gradually increasing reasoning time thresholds, the corresponding maximum state space size, for which none of the BNs exceed the reasoning time, grows much more gradually than the corresponding treewidth.

These results provide a solid basis for our objective to develop algorithms that already bound maximum state space size during BN structure learning, as we will lay out in the next sections.

## 4 BN Learning of Bounded State Space Size

### 4.1 Modified Heuristics

We now describe the modifications we made to the k-MAX and k-greedy heuristics to operate with a bound on the state space.

Now, we give a brief outline of the k-greedy algorithm. Let us assume a treewidth bound of $k$. The algorithm starts from a random ordering of the random variables. *Initialization:* It then initializes the first bag with the first $k+1$ variables. It computes the best DAG over these variables either by an exact method or by an approximate method depending on the value of $k + 1$. *Addition:* Then, the algorithm iteratively adds variables to this DAG with parent sets that maximize local score. While doing so, the algorithm searches through the existing $k$-cliques as potential parent sets. *Termination:* The iteration continues until there are no variables left to add.

To make the k-greedy algorithm work for bounded state space, we first modify the Initialization step. Let $\sigma = v_1, v_2, \ldots, v_n$ be the randomly sampled ordering. Instead of simply picking the first $k$ variables, we now pick the first $p$ variables such that $p$ is the largest integer satisfying the condition $\prod_{i=1}^{p} v_i \leq k$. In the Addition step, we no longer treat all existing $k$-cliques as potential parent sets. Instead, when adding variable $v$, we only consider those sets $S$ for which $\prod_{u \in S \cup \{v\}} \mathrm{ds}(u) \leq k$.

The k-MAX algorithm is similar to the k-greedy algorithm but picks the variables based on a scoring system instead of following a particular variable ordering. The modifications we propose for the k-greedy algorithm can, however, be easily adjusted to work for the k-MAX algorithm by incorporating domain-size checks in place of cardinality checks.

We thus obtain the following algorithms:

**k-greedy and k-MAX** refer to the original two heuristics proposed by Scanagatta et al. [2016, 2018] that return bounded treewidth BNs.
**k-greedy$^{\text{bss}}$ and k-MAX$^{\text{bss}}$** refer to the modified versions of k-greedy and k-MAX with the proposed modifications that return bounded state space BNs.

As a side effect, all these algorithms produce a tree decomposition that witnesses the treewidth or state space bound of the obtained BN.

## 4.2 Local Improvement

In this section, we explain how the local improvement framework of Peruvemba Ramaswamy and Szeider [2021a] can be extended and utilized for the bounded state space BN structure learning problem.

Throughout this section, consider an instance of the bounded state space BN structure learning problem, consisting of a set $V$ of random variables, a score function $f$, and a state space bound $k$. *Initialization:* We first use a heuristic (such as described in Subsection 4.1) to compute an *initial solution* $D$, together with a tree decomposition $\mathcal{T} = (T, \chi)$ of the moralized graph $M(D)$ with maximum state space size $\leq k$. The local improvement step uses the parameter *budget* $B$ which controls the size of the local instance. *Subtree selection:* We select a subtree $S$ of $T$ such that $V_S := \bigcup_{t \in V(S)} \chi(t)$ does not exceed the budget $B$. Our aim is to compute for each $v \in V_S$ a new parent set, optimizing the score of the resulting DAG $D^{\text{new}}$ with $V(D^{\text{new}}) = V$. We define $D_S^{\text{new}}$ as the DAG induced by $V_S$ where $E(D_S^{\text{new}}) = \{ (u, v) \in E(D^{\text{new}}) : \{u, v\} \subseteq V_S \}$. We distinguish between different kinds of nodes:

- $v \in V_S$ a *boundary vertex* if there exists a tree node $t \in V(T) \setminus V(S)$ such that $v \in \chi(t)$;
- $v \in V_S$ is an *internal vertex* if $v$ is not a boundary vertex;
- $v \in V \setminus V_S$ is an *external vertex*.

Two boundary vertices $v, v'$ are *adjacent* if both occur together in some bag outside $S$. In that case we call $\{v, v'\}$ a *virtual edge*. We let $E_{\text{virt}}$ be the set of all virtual edges. The *extended moral graph* $M_{\text{ext}}$ is obtained from $M(D_S^{\text{new}})$ by adding all virtual edges. If $v, v'$ are two adjacent boundary vertices such that $D^{\text{new}}$ contains a directed path from $v'$ to $v$, where all the vertices on the path, except for $v'$ and $v$, are external, then $(v', v)$ is a *virtual arc*. $E_{\text{virt}}^{\rightarrow}$ denotes the set of all virtual arcs.

We call $D_S^{\text{new}}$ a *well-behaved* DAG with respect to DAG $D$ and its tree decomposition $(T, \chi)$ if the following conditions are satisfied:

1. $D_S^{\text{new}}$ is acyclic.
2. For each $v \in V(M_{\text{ext}})$, if $P_{D^{\text{new}}}(v)$ contains external vertices, then there is some $t \in V(T) \setminus V(S)$ such that $P_{D^{\text{new}}}(v) \cup \{v\} \subseteq \chi(t)$.
3. The digraph with vertex set $V(M_{\text{ext}})$ and arc set $E(D_S^{\text{new}}) \cup E_{\text{virt}}^{\rightarrow}$ is acyclic.

Let $\mathcal{T} = (T, \chi)$ be a tree decomposition of DAG $D$. We call $\mathcal{T}^{\text{new}} = (T^{\text{new}}, \chi^{\text{new}})$ a *conservative* tree decomposition of DAG $D^{\text{new}}$ with respect to set $S \subseteq T$ if

1. $\mathcal{T}^{\text{new}}$ is a tree decomposition of DAG $D^{\text{new}}$,
2. $T^{\text{new}}$ can be partitioned into $S^{\text{new}}, T_1, \ldots, T_r$, where $T_1, \ldots, T_r$ are the connected components of $T \setminus V(S)$, and
3. $\chi^{\text{new}}(t) = \chi(t)$ for $t \in \bigcup_{i=1}^{r} V(T_i)$.

**Lemma 1.** *If $D_S^{\text{new}}$ is well-behaved, then $D^{\text{new}}$ is acyclic and $f(D^{\text{new}}) \geq f(D)$.*

*Proof.* The correctness follows from [Peruvemba Ramaswamy and Szeider, 2021a, Theorem 1], as the statement of the lemma is identical except the treewidth constraints. We note that any tree decomposition of such a $D_S^{\text{new}}$ is conservative with respect to $S$. □

Introducing a state space bound to Lemma 1, we obtain the following theorem.

**Theorem 1.** *If $D_S^{\text{new}}$ is well-behaved and admits a tree decomposition with* msss *at most $k$, then*

1. *$D^{\text{new}}$ is acyclic,*
2. *the score of $D^{\text{new}}$ is at least the score of $D$, and*
3. *$\text{msss}(D^{\text{new}}) \leq k$.*

*Proof.* The first two properties follow from Lemma 1. Now, let $\mathcal{T} = (T, \chi)$ be the tree decomposition of DAG $D$, $\mathcal{T}^{\text{new}} = (T^{\text{new}}, \chi^{\text{new}})$ be the tree decomposition of DAG $D^{\text{new}}$ and $\mathcal{S}^{\text{new}} = (S^{\text{new}}, \chi_S^{\text{new}})$

be the tree decomposition of $D_S^{\text{new}}$. Since $\mathcal{S}^{\text{new}}$ is conservative with respect to $S$ (from Lemma 1), we can express $\chi^{\text{new}}$ as $\chi_S^{\text{new}}(t)$ if $t \in S$ and $\chi(t)$ otherwise. In order to prove the third property, we need to bound the state space size of the bags of $\mathcal{T}^{\text{new}}$, i.e., we need to bound

$$\max_{t \in T^{\text{new}}} \prod_{v \in \chi^{\text{new}}(t)} \text{ds}(v) = \max \left( \max_{t \in S} \prod_{v \in \chi_S^{\text{new}}(t)} \text{ds}(v), \max_{t \in T \setminus V(S)} \prod_{v \in \chi(t)} \text{ds}(v) \right)$$

$$\leq \max \left( k, \max_{t \in T} \prod_{v \in \chi(t)} \text{ds}(v) \right) \qquad [\text{msss}(D_S^{\text{new}}) \leq k]$$

$$\leq \max(k, k) \qquad [\text{msss}(D) \leq k].$$

Thus, the maximum state space size of $D^{\text{new}}$ is at most $k$. $\qquad \square$

### 4.3 MaxSAT Encoding

We now describe how we construct the (weighted, partial) MaxSAT instance that encodes the conditions required on the local instance $S$. The instance is a propositional formula in conjunctive normal form (CNF), with hard clauses and soft clauses; each soft clause has a weight. The MaxSAT solver tries to find a truth assignment that satisfies all the hard clauses and maximizes the sum of weights of satisfied soft clauses. We take as input the local instance $S$, the set of virtual edges $E_{\text{virt}}$, the set of virtual arcs $E_{\text{virt}}^{\rightarrow}$, and the bound $k$ on the maximum state space size and produce a MaxSAT instance $\Phi_S$. We reuse parts of the encoding proposed by Peruvemba Ramaswamy and Szeider [2021a] (we will refer to it as the *BN-SLIM encoding*), extending an encoding proposed by Samer and Veith [2009] for capturing the tree decompositions.

Let $n = |S|$ denote the size of the subinstance. We now reiterate the hard clauses from the BN-SLIM encoding and refer to the conjunction of these clauses as the formula $\Phi_S'$. For an explanation of the semantics of these variables and clauses, we refer to the source [Peruvemba Ramaswamy and Szeider, 2021a].

$$\left. \begin{array}{r} (\text{acyc}_{u,v}^* \wedge \text{acyc}_{v,w}^*) \to \text{acyc}_{u,w}^* \\ (\text{ord}_{u,v}^* \wedge \text{ord}_{v,w}^*) \to \text{ord}_{u,w}^* \end{array} \right\} \quad \text{for distinct } u, v, w \in S.$$

$$\neg \text{arc}_{v,v} \qquad \text{for } v \in S.$$

$$\sum\nolimits_{P \in \mathcal{P}_v} \text{par}_v^P = 1 \qquad \text{for } v \in S.$$

$$\text{par}_v^P \to \text{acyc}_{u,v} \qquad \text{for } v \in S, P \in \mathcal{P}_v, \text{ and } u \in P.$$

$$\left. \begin{array}{r} (\text{par}_v^P \wedge \text{ord}_{u,v}) \to \text{arc}_{u,v} \\ (\text{par}_v^P \wedge \text{ord}_{v,u}) \to \text{arc}_{v,u} \end{array} \right\} \quad \text{for } v \in S, P \in \mathcal{P}_v, \text{ and } u \in P.$$

$$\left. \begin{array}{r} (\text{par}_v^P \wedge \text{ord}_{u,w}) \to \text{arc}_{u,w} \\ (\text{par}_v^P \wedge \text{ord}_{w,u}) \to \text{arc}_{w,u} \end{array} \right\} \quad \text{for } v \in S, P \in \mathcal{P}_v, \text{ and } u, w \in P.$$

$$\left. \begin{array}{r} (\text{arc}_{u,v} \wedge \text{arc}_{u,w} \wedge \text{ord}_{v,w}) \to \text{arc}_{v,w} \\ (\text{arc}_{u,v} \wedge \text{arc}_{u,w} \wedge \text{ord}_{w,v}) \to \text{arc}_{w,v} \end{array} \right\} \quad \text{for } u, v, w \in S.$$

$$\neg \text{arc}_{u,v} \vee \neg \text{arc}_{v,u} \qquad \text{for } u, v \in S.$$

$$\text{ord}_{u,v}^* \to \text{arc}_{u,v} \wedge \text{ord}_{v,u}^* \to \text{arc}_{v,u} \qquad \text{for } \{u, v\} \in E_{\text{virt}}.$$

$$\text{par}_v^P \to \text{acyc}_{u,v}^* \qquad \text{for } v \in S, P \in \mathcal{P}_v, \text{ and } (u, v) \in A_{\text{virt}}^{\rightarrow}(v, P).$$

We also add the following soft clauses to $\Phi_S'$, setting $f_P'(v) = f_P(v) - f_\emptyset(v)$ as their weight

$$(\text{par}_v^P) : \text{weight } f_P'(v) \quad \text{for } v \in S, P \in \mathcal{P}_v.$$

The weight of a solution to $\Phi_S'$ is given by the sum of weights of the satisfied soft clauses. Let $W = \sum_{v \in S} f'(v, P_D(v))$ be the core of the unmodified local instance $S$.

**Lemma 2.** $\Phi_S'$ *admits a solution of weight* $W^{\text{new}}$ *if and only if there exists a well-behaved DAG* $D_S^{\text{new}}$ *with respect to $D$, such that* $f(D^{\text{new}}) - f(D) = W^{\text{new}} - W$.

*Proof.* We can use the proof of [Peruvemba Ramaswamy and Szeider, 2021a, Theorem 2] to establish the lemma, since the only difference the clauses that upper bound treewidth of $D_S^{\text{new}}$. Therefore $D_S^{\text{new}}$

is still acyclic. Further, an assignment that corresponds to setting $D_S^{\text{new}} = S_S$ achieves the lower bound on the score. □

## 4.4 BDD-based Counter

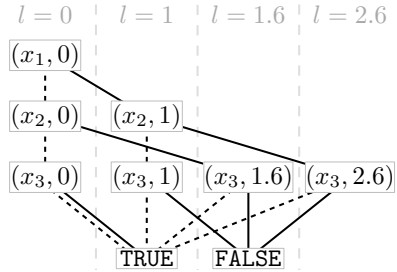

Figure 3: Example BDD constructed for three variables $v_1, v_2, v_3$ with domain sizes $2, 3, 4$, respectively, and with $k = 6$. The logarithms of the domain sizes are $1, 1.6, 2$, respectively. The solid and dashed edges represent the 'if' and 'else' arcs respectively. The path $(x_1, 0) \text{---} (x_2, 1) \text{---} (x_3, 2.6) \text{- -} \texttt{TRUE}$ represents the bag containing $x_1$ and $x_2$ with a state space size of $6 \leq k$, hence reaching $\texttt{TRUE}$. The path $(x_1, 0) \text{---} (x_2, 1) \text{- -} (x_3, 1) \text{---} \texttt{FALSE}$ represents the bag containing $x_1$ and $x_3$ with a state space size of $8 > k$, hence reaching $\texttt{FALSE}$. Note that both these bags have cardinality 2 and hence are treated the same when it comes to treewidth.

In this section, we elaborate the counting mechanism that we use to encode the condition $\sum_{w \in S, w \neq v, \text{arc}_{v,w}} \log(\text{ds}(w)) \leq \log(k)$ for $v \in S$. The technique we use was first proposed by Eén and Sörensson [2006]. Intuitively speaking, we construct an (Ordered) Binary Decision Diagram (BDD), where following a path from an input node to a terminal node corresponds to summing up weights. Each layer of nodes in the BDD is associated with a variable $v \in S$, and there are two outgoing edges from each node in the BDD corresponding to the existence and absence of $\text{arc}_{v,w}$. More formally, let us assume that the set $S$ consists of the variables $v_1, \ldots, v_m$, where $m := |S|$. We construct a BDD, which is a directed graph with nodes of the form $(x_i, l)$ where $i \in \{1, d \ldots m\}$ is an integer, and $l \in [0, k]$ is a real number. Here, $x_i$ signifies that we branch on variable $v_i$ next, and $l$ denotes the level of the current sum of weights. Additionally, there are two special terminals (sink nodes) – $\texttt{TRUE}$ and $\texttt{FALSE}$. From each node $(x_i, l)$ there are two outgoing arcs – the 'if' arc connecting it to the node $(x_{i+1}, l + \log(\text{ds}(v_i)))$, and the 'else' arc connecting it to the node $(x_{i+1}, l)$. In case $l + \log(\text{ds}(v_i)) > k$, we instead connect it to the $\texttt{FALSE}$ terminal. Finally, for a node of the form $(x_m, l)$, we connect it to the $\texttt{TRUE}$ terminal iff $l + \log(\text{ds}(v_m)) \leq k$. We denote by $\text{if}(x, l)$ and $\text{else}(x, l)$, the nodes connected to $(x, l)$ via the 'if' and 'else' arcs, respectively. We refer to Figure 3 for an example. There are at most $2^m + 1$ nodes in the BDD.

For each variable $v \in S$, we have one such BDD to ensure that the product of the domain sizes of the endpoints of the outgoing arcs does not exceed the bound $k$. It is straightforward to express this BDD in the form of a CNF formula $\mathcal{B}_v$. We introduce a variable $\text{bdd}_{w,l}^v$ for each $w \in S \setminus \{v\}$ and each level $l$ in the BDD. We then add the following clauses to $\mathcal{B}^v$

$$\left. \begin{array}{l} (\text{bdd}_{w,l}^v \wedge \text{arc}_{v,w}) \to \text{bdd}_{\text{if}(w,l)}^v \\ (\text{bdd}_{w,l}^v \wedge \neg\text{arc}_{v,w}) \to \text{bdd}_{\text{else}(w,l)}^v \end{array} \right\} \quad \text{for } (w, l) \in V(\mathcal{B}_v).$$

Next, we add the clause $\neg\text{bdd}_{\texttt{FALSE}}^v$ to $\mathcal{B}^v$ that falsifies $\mathcal{B}^v$ if the $\texttt{FALSE}$ sink node is reached, i.e., if the outgoing arcs of variable $v$ violate the bound. Finally, we conjoin all the formulas $\mathcal{B}^v$ for $v \in S$ with the formula $\Phi'_S$ to obtain our final formula $\Phi_S$, giving us the following theorem.

**Theorem 2.** $\Phi_S$ *admits a solution of weight* $K^{\text{new}}$ *if and only if there exists a well-behaved DAG* $D_S^{\text{new}}$ *with respect to* $D$, *such that* $f(D^{\text{new}}) - f(D) = K^{\text{new}} - K$ *and* $\text{msss}(D_S^{\text{new}}) \leq k$.

*Proof.* By construction of the BDD, we see that $\mathcal{B}^v$ is falsified if and only if the outgoing arcs from variable $v$ result in the state space size of that bag exceeding $k$. Thus, $\bigwedge_{v \in S} \mathcal{B}^v$ is satisfiable if and only if $\text{msss}(D_S^{\text{new}}) \leq k$. Combining this with Lemma 2, we obtain the desired result. □

Finally, we would like to draw attention to the fact that since only the local part is encoded into a MaxSAT formula, increasing the total number of variables does not directly affect the solving time for each individual subinstance, and consequently, the time required for each individual improvement. Thus the overhead of the BDDs doesn't increase when the total number of variables in the network increases. This is a crucial strength of the SLIM approach in general, i.e., it scales well relative to the total number of variables, and the runtime has a stronger dependence on the budget.

# 5 Experiments

## 5.1 Setup

We tested the various proposed methods on 4-core Intel Xeon E5540 2.53 GHz CPU (internal cluster), with each process having access to 8GB RAM. The k-greedy and k-MAX algorithms are available as a part of the BLIP package [Scanagatta, 2015] implemented in Java. We modified and extended these to obtain the implementations for k-greedy$^{bss}$ and k-MAX$^{bss}$. We implemented BN-SLIM$^{bss}$in Python making use of the NetworkX library [Hagberg et al., 2008]. We used UWrMaxSat[1] as the MaxSAT-solver, because of its reliable response to timeouts. For evaluating the reasoning time we used the Merlin package by Radu Marinescu.[2] We provide the source code as a public github repository [Peruvemba Ramaswamy and Szeider, 2021b].

We tested the algorithms on a subset of the `bnlearn` repository.[3] These networks are commonly used as benchmarks in the literature. Out of the 22 networks available in the repository, we only consider the 16 networks that contain non-binary variables. These networks range in size from 6 to 1041 random variables. Due to the smaller networks' behavior being susceptible to random noise, we focus more on the larger networks.

## 5.2 Method

Next, we describe the method used to evaluate the performance of the BN structure learning algorithms enumerated in Section 4.1. We run the algorithms for a total time of 90 minutes and record the reasoning time and score at the end. We denote by BN-SLIM$^{bss}$(X) the algorithm composed of running the heuristic X for 30 minutes and then running the bounded state space local improvement algorithm on top of the heuristic solution for another 60 minutes. We run the algorithms with multiple bounds and multiple random seeds. Finally, for each instance, we visualize the distribution of scores and reasoning times on a scatter plot to capture the tradeoff achieved between the two metrics. We set k-greedy and k-MAX as the *baseline* algorithms and always compare the newly proposed algorithms against these baseline algorithms. The baseline algorithms represent the state-of-the-art bounded treewidth BN structure learning methods while the heuristics represent the bounded state space based BN structure learning methods.

## 5.3 Results

We performed some preliminary tests to determine the most promising bounded state space (bss) methods for comparison against the baseline of tw-methods k-MAX and k-greedy. On the basis of these tests, we choose BN-SLIM$^{bss}$(k-MAX$^{bss}$) as the bss-method. In Figure 4 we see the scatter plots for a representative subset of instances. We include the complete set of plots in the supplementary material. The plots are presented in pairs with the left subplot depicting the tw-methods' performance and the right subplot depicting BN-SLIM$^{bss}$(k-MAX$^{bss}$)'s performance.

In general, from Figure 4, we observe that BN-SLIM$^{bss}$(k-MAX$^{bss}$) achieves much faster reasoning times at the expense of slightly worse scores in some cases. Whereas in some cases, BN-SLIM$^{bss}$(k-MAX$^{bss}$) manages to match the score while reducing the reasoning time significantly. In most cases, we observe an reduction by an order of magnitude. Another point worth noting is that the clustering of the BNs output by BN-SLIM$^{bss}$(k-MAX$^{bss}$) along the reasoning time axis is much tighter than with the tw-methods. This highlights the reliability aspect of the bss-methods.

We also observe that the bss-method BN-SLIM$^{bss}$(k-MAX$^{bss}$) allows us to expand the search space much more carefully and predictably. For instance, consider a bound on the maximum state space size of $5 \times 10^5$. A reasonable equivalent bound on the treewidth would be $\log_2(5 \times 10^5) \approx 19$. However, networks with treewidth 19 can be expected to have much worse inference speeds as compared to networks with msss $\leq 5 \times 10^5$.

This confirms our hypothesis that maximum state space size is a much better estimator of reasoning time as compared to treewidth, and that one can construct algorithms like BN-SLIM$^{bss}$(k-MAX$^{bss}$) to learn such fast-inference BNs.

---

[1] `https://maxsat-evaluations.github.io/2019/descriptions.html`
[2] `https://github.com/radum2275/merlin`
[3] `https://www.bnlearn.com/bnrepository/`

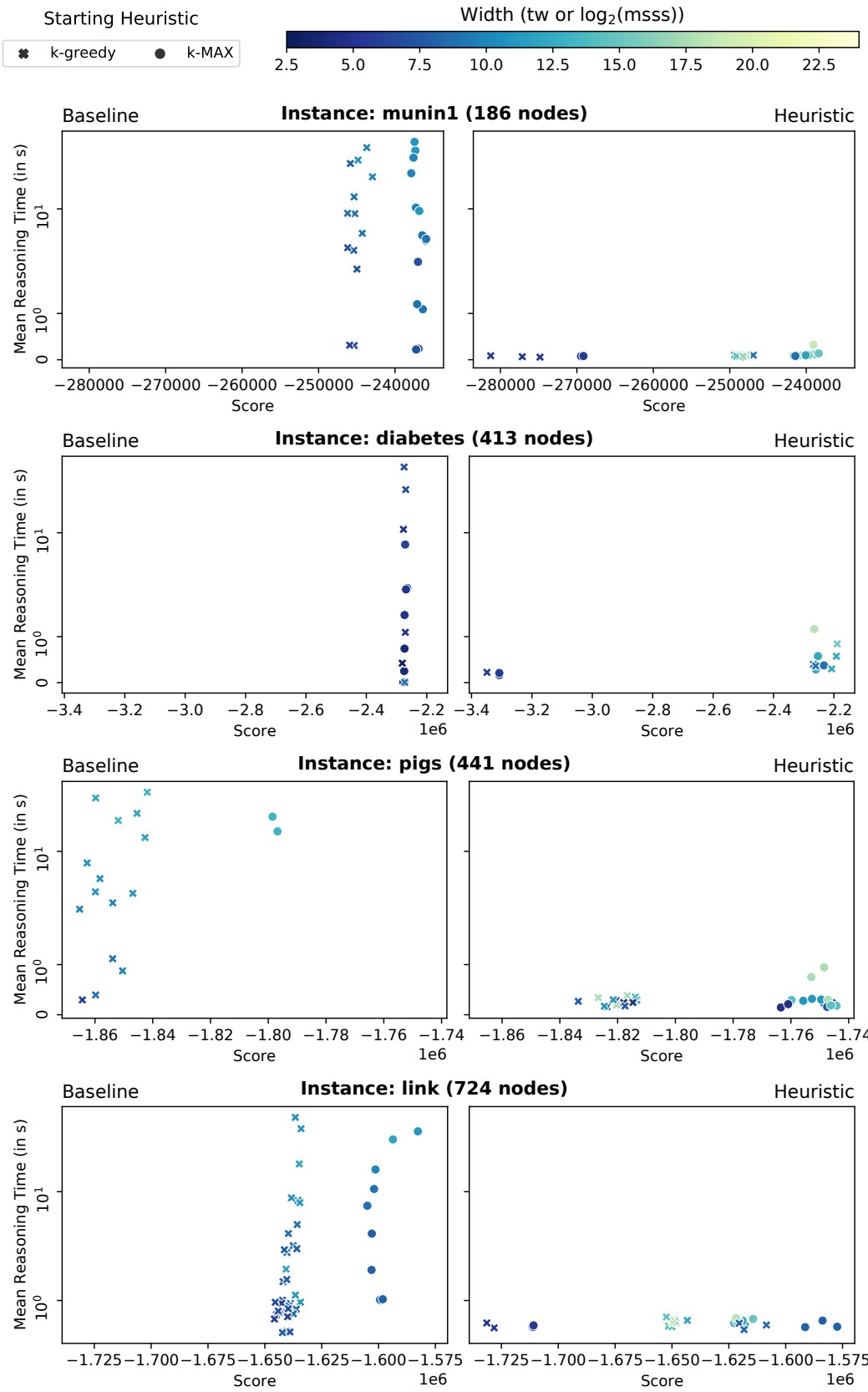

Figure 4: Scatter plots

# 6  Conclusion

We have introduced the concept of bounded state space BN structure learning, devised it theoretically, implemented it, and tested it rigorously on a set of real-world data sets. We compared the new approach with state-of-the-art bounded treewidth BN structure learning algorithms on a two-dimensional setting to see the tradeoff between inference speed and data fitting. Our results show that the new approach indeed provides overall better and more reliable results. In some cases, the advantage of bounded state space over bounded treewidth methods is significant.

**Ethical Impact**  BNs are widely used in decision-making loops. We suspect our paper doesn't create any new negative social impacts than those that were already present. However, it could lead to over-dependence on BNs, which has its risks. In its current state, there are many areas where BNs cannot replace a domain expert. We propose methods to speed up the inference on BNs, which does not significantly change their use case.

## Acknowledgments and Disclosure of Funding

The authors acknowledge the support by the FWF (projects P32441 and W1255) and by the WWTF (project ICT19-065).

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
