# OpenReview forum: "Learning Fast-Inference Bayesian Networks"
_NeurIPS.cc/2021/Conference — NeurIPS 2021 Poster_

### Official Review · Reviewer_mMbj · 2021-07-12

**Rating:** 7
**Confidence:** 4

**Summary:**

A method for learning BNs where the size of the joint state space of a
bag in a tree-decomposition is bounded. It is argued (convincingly)
that this is a more useful quantity to bound than simple treewidth,
and empirical results produced to back this up. Learning has two
phases: an initial heuristic, followed by "local improvement", the
latter developed from earlier work. A (weighted partial) MAXSAT
encoding is presented for the local improvement phase which is an
extension from earlier work. A key contribution is a method for
encoding the constraint on the size of the joint state space: move to
logs, replace product with sums, and a BDD to encode the necessary summation.

**Limitations And Societal Impact:**

The authors consider this.

**Main Review:**

The methods are mainly not-too-big alterations and additions to
existing methods. But I think this is a good thing. The authors know
the literature and have thus saved themselves from (re)inventing some
(possibly inferior) methods.

This is a clearly written paper. We have some useful theoretical
results, but the key issues are: (1) are we using the right measure of
"responsiveness"? and (2) do we learn the "right" networks?

Re (1) Section 3.1 provides evidence for a "yes" answer. Re (2),
naturally there could be a trade-off between fit-to-data and
responsiveness. Fig 4 gives us a reasonable idea of when there is such
a trade-off. I think the authors have provided sufficient evidence
that that the presented method produces the right sorts of BNs.

Not a big issue, but I don't think "responsive" is the right word to
use for BNs which admit speedy query answering, since it does not
convey that the response (to a query) comes back quickly.

**Time Spent Reviewing:**

2

---

> ### Author Response · Authors · 2021-08-09
> **Response to Reviewer mMbj**
>
> Thank you for your feedback. As for your point about the term “responsive”, we will try to find a more appropriate term for the final paper.

---

> > ### Comment · Reviewer_mMbj · 2021-08-30
> > **response to author response**
> >
> > OK, I look forward to seeing which new word you choose!

---

### Official Review · Reviewer_f3K7 · 2021-07-15

**Rating:** 6
**Confidence:** 4

**Summary:**

This is a paper about the structural learning of Bayesian networks. Analogously to existing methods trying to learning models with bounded treewidth, the authors here are interested in learning models with bounded "state space". Of course the two descriptors differ for non-binary models, the second providing a more precise indicator of the time needed for inferences.

The approach achieves that by a (simple) modification of the algorithms proposed by Scanagatta et al. for bounded treewidth.
This is followed by a post-processing step which adapts a similar approach by Ramaswamy and Szeider (AAAI 2021) for bounded treewidth to the bounded state space case by a MaxSAT reduction. Experiments are showing how this approach might give models producing faster inferences, but not necessarily higher scores.

**Ethical Concerns:**

Nothing.

**Limitations And Societal Impact:**

Nothing.

**Main Review:**

The idea is clear and reasonable. Yet, I see two critical points:

(i) the reason for focusing on bounded state spaces (and not on bounded treewidth) should be better advocated. I can imagine significant differences for the reasoning time only when variables with a huge number of states are present.
(ii) compared to Ramaswamy and Szeider (AAAI 2021) the contribution sounds a bit incremental, being basically an adaptation of what has been done in that paper for bounded treewidth to the bounded state space.
(iii) the experimental results are not well organized, a synthetic (quantitative) description is missing

*** I eventually decided to revise my score after the rebuttal. ***



**Time Spent Reviewing:**

1

---

> ### Author Response · Authors · 2021-08-09
> **Response to Reviewer f3K7**
>
> Thank you for your feedback, please find our response below, along with the relevant quotes from the review.
>
> ---
> > I can imagine significant differences for the reasoning time only when variables with a huge number of states are present.
>
> As mentioned by another reviewer, it is fairly common to encounter non-binary variables in real-world data.
> Moreover, during our preliminary analysis, we noticed that even having variables with domain sizes as low as 4 was sufficient to impact the reasoning time.
> This agrees with the fact that the reasoning time has an exponential dependence on the domain sizes.
> We will add a discussion about this point to further emphasize the relevance and significance of the state space sizes.
>
> For example, some of the networks learned for the instances `alarm` and `hepar2` exhibited relatively high reasoning times despite having small number of variables and small domain sizes
>
> | Instance | # variables | Max domain size | Mean domain size |
> |:---------|------------:|----------------:|-----------------:|
> | `alarm`  |          37 |               4 |             2.84 |
> | `hepar2` |          70 |               4 |             2.31 |
>
> ---
> > (ii) compared to Ramaswamy and Szeider (AAAI 2021) the contribution sounds a bit incremental
>
> We agree that it’s a bit incremental in the sense that we build upon a well-established methodology and the work by Ramaswamy and Szeider (AAAI 2021). However the delta is significant as it includes the following main contributions
> * Theorem 1
> * BDD-based workflow for capturing bounded state space size
> * Non-trivial implementation effort of modifying heuristics ($\text{k-MAX}^\text{bss}$ and $\text{k-greedy}^\text{bss}$)
> * Carrying out a rigorous experimental investigation (over 45,000 runs) on the effect of optimizing for bounded state space size instead of bounded treewidth
>
> ---
> > (iii) the experimental results are not well organized, a synthetic (quantitative) description is missing
>
> Due to the nature of the problem at hand, it might be impossible to improve both the score of the network and the reasoning time compared to the solution provided by the existing heuristics. Hence, we set our primary objective as improving the reasoning time, with the secondary objective being to mitigate the drop in the score. As a result, we found it very difficult to compare and analyze the performance of the different methods using just scalars and 2D plots. In our opinion, the plots we finally ended up with, which we included in the paper, are the best (albeit limited) representation of the performance comparison.

---

### Official Review · Reviewer_tk5p · 2021-07-16

**Rating:** 6
**Confidence:** 3

**Summary:**

A structure learning method for Bayesian networks. The main contribution is the use of state space to bounds in structure learning rather than the more well established treewidth bounds to improve scalability an speed of inference.

**Limitations And Societal Impact:**

Addressed by the authors. Since the main contribution is improving scalability of existing models, I do not foresee negative societal impact.

**Main Review:**

The paper proposes a new structure learning algorithm for Bayesian nets. The main innovation is to use a bound on the state space as compared to tree width bounds that are typically used in structure learning. In Bayesian nets with non-binary variable domains, the state space bound can be a  much more effective bound for learning compared to tree width. Two known heuristics are augmented with the state space bound. A recently developed approach is adapted to include the state space bound. To guarantee the constraints on the state space, a maxSAT encoding is constructed. A counting method using binary decision diagrams is constructed where a solution ensures that the state space bound can be met.

Experiments are performed on standard benchmarks for structure learning and improvements are shown over tree width based structure learning in terms of mean reasoning time, i.e. using the state space bounds we can learn Bayesian nets where inference is faster.

The paper has an interesting new approach to use state space bounds quite different from the tree width bounds commonly used. The case for non-binary domains seem relevant since a lot of real world problems have non binary domains and seems like tree width is a worse bound for such cases, so the proposed approach seems significant. One weakness is perhaps the paper builds a lot on recent results by Ramaswamy and Szeider, so the main novel contribution seems to be theorem 1 and the BDD encoding since the maxsat encoding seems to be known already. Maybe explicitly specifying the new contribution will help understand the significance of the proposed method better.

The experiments do show a consistently significant reduction in reasoning time compared to state of the art implemented in the BLIP package over all the benchmarks, so the experiments do show the benefit of the approach. I am wondering if scalability of structure learning is an issue for large number of variables since there is an overhead of using BDDs for each variable. Overall, the paper though seems to make good contributions with good empirical results to back them up. Some comments about the significance of the contributions beyond the Ramaswamy and Szeider 2021 work will help.


**Time Spent Reviewing:**

4

---

> ### Author Response · Authors · 2021-08-09
> **Response to Reviewer tk5p**
>
> Thank you for your feedback, please find our response below, along with the relevant quotes from the review.
>
> ---
> > Maybe explicitly specifying the new contribution will help understand the significance of the proposed method better.
>
> Here is a list with our main contributions
> * Theorem 1
> * BDD-based workflow for capturing bounded state space size
> * Non-trivial implementation effort of modifying heuristics ($\text{k-MAX}^\text{bss}$ and $\text{k-greedy}^\text{bss}$)
> * Carrying out a rigorous experimental investigation (over 45,000 runs) on the effect of optimizing for bounded state space size instead of bounded treewidth
>
> ---
> > I am wondering if scalability of structure learning is an issue for large number of variables since there is an overhead of using BDDs for each variable.
>
> Since only the local part is encoded into a MaxSAT formula, increasing the total number of variables doesn’t directly affect the rate of improvements i.e., the time required for each individual improvement. Thus the overhead of the BDDs doesn’t become any worse if the total number of variables in the network increases.
> This is, in fact, a key strength of the BN-SLIM approach in general, i.e., it scales well relative to the total number of variables and the runtime has a stronger dependence on the budget (size of local subinstance).

---

### Official Review · Reviewer_9g4D · 2021-07-19

**Rating:** 6
**Confidence:** 3

**Summary:**

This work combines heuristic Bayesian network structure learning algorithms with a Max-SAT-based local improvement method to provide more responsive than BNs provided by treewidth-based methods. Some efforts are devoted to generate fast methods when learning a Bayesian network from data, but it is also important to provide the ‘best possible structure’. Once the network is created (learned or elicited), the use will imply generally multiple inference operations which is the usual way to reason in BNs. This work aims at generating responsive BNs, that is, able to admit fast probabilistic reasoning. In order to allow tractable inference we need to learn Bayesian networks with a bounded-treewidth structure, being then treewidth value of extreme importance to determine the speeding reasoning. Let us remind that the treewidth of a triangulated graph is the size of its largest clique/bag minus one, so it will determine the largest message to be sent, if probability propagation is used for inference. However, state space is also important, as not all the variables present the same number of states, a binary variable will introduce less potential combinations than one with 5 states for instance. The state space of a clique/bag is the product of the domain sizes of the variables it contains. There exist evidence, supported by the initial experiments, which shows that the largest state space size of all bags can be even more decisive on the reasoning time than the treewidth, which seems natural as it is more informed about the dimension of the bag.

Authors’ proposal is heavily based on proposed the hybrid approach BN-SLIM, which improves the score of a heuristically computed BN by multiple applications of a MaxSAT-based exact method. They compare their methodology using  bounded state space (bss) with other baseline methods establishing bounded-treewidth strategies, namely k-greedy and k-MAX. Results indicate that bss methods perform better and faster, both BN-SLIMbss and k-MAXbss allows expanding the search space more carefully and predictably.

**Limitations And Societal Impact:**

It is properly addressed.

**Main Review:**

In my opinion the work is original enough, even though it is greatly based on an existing work, it makes an important contribution when using the bss approach.

The quality is correct. Some parts may assume too specific knowledge but it is possible to be followed by a familiarised reader.

There are a few typos (*) but in general the paper is well written. The structure is a little bit unbalanced, as the results are presented too quickly and without a thorough analysis. Even though the supplementary material provides further plots, I would be happy to see in the paper itself a deeper analysis on the results, maybe using some other objective scores to compare the methods.

The results are of moderate significance, I personally find that exploring the bss alternative is clearly promising.

(*) I will list those detected:

- page 4: consisting pf …
- page 5: Our aim is compute
- page 8: networks' behvaior

I like the way in which section 3 presents the two possibilities, and I think section 4 seems correct, although I did not verify every mathematical detail. As indicated before, the experimental side is a little poor, probably due to space restrictions, but I would personally prefer to keep some proofs as supplementary material and find a more in-depth comparison with a broader analysis in the published paper.

In general I believe it is a paper valuable for discussion and offering helpful and promising ideas.


**Time Spent Reviewing:**

2

---

> ### Author Response · Authors · 2021-08-09
> **Response to Reviewer 9g4D**
>
> Thank you for your feedback, please find our response below, along with the relevant quotes from the review.
>
> ---
> > I would be happy to see in the paper itself a deeper analysis on the results, maybe using some other objective scores to compare the methods.
>
> Due to the specific nature of the problem at hand, we found it tricky to compare, aggregate and visualize the different parameters involved and finally landed on the plots that we currently have included in the paper. That being said, we will try to exhibit some of the results in the form of tables and come up with alternative visual means to resolve this concern.
>
> ---
> > (*) I will list those detected:
>
> Thank you for pointing out the typos.

---

### Decision · Program_Chairs · 2021-09-27

**Decision:**

Accept (Poster)

**Comment:**

This manuscript proposes to use a measure based on the max. "bag" size (product of cardinalities) when learning Bayesian networks, instead of treewidth. From a theoretical perspective, this is unimportant: results of hardness hold when all variables have the same cardinality, and one can easily convert nets if needed to/from that. However, the practical results of using the bag size (which is obviously a better measure than treewidth of the graph, no doubt of that, everybody should know it) can be interesting and useful. The novelty and theoretical contribution are not so significant, but on the other hand it is fair to say that someone needs come forward and argue that we should be using bag size instead of treewidth and make that claim work properly.